# Medical Radiology: Current Progress

**DOI:** 10.3390/diagnostics13142439

**Published:** 2023-07-21

**Authors:** Alessia Pepe, Filippo Crimì, Federica Vernuccio, Giulio Cabrelle, Amalia Lupi, Chiara Zanon, Sebastiano Gambato, Anna Perazzolo, Emilio Quaia

**Affiliations:** 1Institute of Radiology, University Hospital of Padua—DIMED, Padua University Hospital, University of Padua, 35122 Padua, Italy; 2Department of Radiology, University Hospital of Padua, 35128 Padua, Italy; 3Institute of Radiology, Department of Medicine, Azienda Ospedaliero-Universitaria Santa Maria della Misericordia, University of Udine, 33100 Udine, Italy

**Keywords:** cardiovascular imaging, rectal imaging, liver imaging, computed tomography, magnetic resonance imaging, ultrasound

## Abstract

Recently, medical radiology has undergone significant improvements in patient management due to advancements in image acquisition by the last generation of machines, data processing, and the integration of artificial intelligence. In this way, cardiovascular imaging is one of the fastest-growing radiological subspecialties. In this study, a compressive review was focused on addressing how and why CT and MR have gained a I class indication in most cardiovascular diseases, and the potential impact of tissue and functional characterization by CT photon counting, quantitative MR mapping, and 4-D flow. Regarding rectal imaging, advances in cancer imaging using diffusion-weighted MRI sequences for identifying residual disease after neoadjuvant chemoradiotherapy and [18F] FDG PET/MRI were provided for high-resolution anatomical and functional data in oncological patients. The results present a large overview of the approach to the imaging of diffuse and focal liver diseases by US elastography, contrast-enhanced US, quantitative MRI, and CT for patient risk stratification. Italy is currently riding the wave of these improvements. The development of large networks will be crucial to create high-quality databases for patient-centered precision medicine using artificial intelligence. Dedicated radiologists with specific training and a close relationship with the referring clinicians will be essential human factors.

## 1. Cardiac Imaging

Cardiovascular diseases constitute the leading cause of global mortality and are a major contributor to reduced quality of life. The diagnosis and management of cardiovascular diseases are heavily reliant on diagnostic imaging. Since 1920, when the early days of cardiac imaging relied on fluoroscopy, orthodiagraphy, and radiography, the diagnosis and treatment of acquired and congenital heart disease (CHD) have been progressively revolutionized by new technologies, such as transthoracic echocardiography (TTE), stress echo (SE), transesophageal echocardiography (TEE), and nuclear myocardial perfusion imaging (NMPI). In recent years, the rates of use of TTE, SE, and TEE have decreased by 3% since 2010, while MPI has decreased by 36%. These data probably also reflect the advances in cardiac CT and MR technology that have come to the forefront during the same time period. The use of cardiac computed tomography (CCT) has increased by 84%, and cardiac magnetic resonance imaging (CMR) has increased by 125% [1].

### 1.1. Computed Tomography

In the early 1970s, the first cardiac computed tomography (CCT) was performed. Considering motion as the main factor that predominantly affected image quality in cardiac imaging, through the implementation of electrocardiographically (ECG) gated CCT, it was possible to assess a wide range of heart diseases, such as left ventricular aneurysms, the enhancement of scars, thrombus, and aortic dissection. Since 1998, advances in multi–detector row CT technology with shorter acquisition times and tailored contrast injection protocols have let CCT to evaluate coronary anatomy and coronary artery disease (CAD). Currently, there are technologies that are oriented toward technical development into a heavily time-resolved equipment (e.g., dual-source CT with 256/384 slice), while others are more focused on increasing the width of the detector (e.g., 256/320/640-slice CT) [2]. Moreover, with the advent of the new photon-counting CT, coronary CCT has demonstrated improved image quality and diagnostic confidence in humans, also orienting toward a tissue characterization that has always been a disadvantage in comparison to CMR [3]. Today, in Italy, the number of imaging centers that perform CCT investigations is constantly growing, as is the number of patients that undergo the test [4].

In Italy, as all over the world, the principal indications for CCT are the exclusion of obstructive CAD in symptomatic patients of intermediate clinical likelihood (class I B) with a high probability of good image quality to improve their referral for angiography thanks to the high negative prediction value of the technique, and the calcium score assessment in asymptomatic individuals as a risk modifier in the cardiovascular risk assessment (class IIb B) [5] (Figure 1).

In the emergency room setting, the request for urgent CCT in patients with acute chest pain is demonstrating a growing trend (Figure 2) and increasingly more hospitals have machines and personnel skilled to perform the “CT triple rule-out protocol” for obstructive CAD, aortic dissection, and pulmonary embolism.

On the other hand, given the “CCT-based opportunity” for a stronger preventive treatment in patients with non-obstructive CAD and high-risk plaques, there has been an increase in CCT requests in asymptomatic patients with CAD risk factors of low-intermediate likelihood as a screening tool [6].

CCT has also been increasingly requested as a roadmap before minimally invasive interventions: transcatheter valve repair/replacement, prosthetic valve assessment in suspected dysfunction or endocarditis, or pre-left atrial ablation [6].

The optimal balance between image quality and radiation dose is fundamental in CCT imaging, not only in the scan acquisition protocols, which are increasingly tailored to the clinical query, but also in image analysis. Post-processing requires a standardized execution with a chosen software package to make sure consistent and accurate measurements are obtained [6]. Post-processing software is being constantly developed in the direction of progressive automation and is able to quantify plaque stenosis severity and plaque burden [7]. Newer techniques include semiautomated volumetric quantification of the non-calcified, low-attenuation coronary plaque component (<30 HU being the most frequently applied cutoff), which together with the presence of positive remodeling and spotty calcifications, can improve the prediction of myocardial infarction in patients with stable chest pain [8]. Moreover, the peri-coronary-fat-attenuation-index (pFAI-cut off > −70.1 Hounsfield Unit) could provide a quantitative measurement of coronary inflammation as an indicator of major cardiac adverse events [9]. Various software that are able to perform the abovementioned quantifications are still committed in clinical research, but their large-scale use will likely spread as further data become available.

Recently, positron emission tomography (PET) with newly adopted radiotracers (^18^F-Sodium Fluoride) has provided unique insights into coronary atheroma activity, acting as a powerful independent predictor of myocardial infarctions [10].

The need to standardize the report and to guide the next steps in improving patient management has driven the efforts of radiologic societies towards the creation of a structured reporting and data system, in particular, the Coronary Artery Disease Reporting and Data System (CAD-RADS). The updated 2022 CAD-RADS (2.0) improved the initial reporting system for CCTA by considering new technical developments in Cardiac CT, including the assessment of lesion-specific ischemia using CT fractional-flow-reserve (CT-FFR) or myocardial CT perfusion (CTP) [7]. The use of a state-of-the-art CCT with a 16-cm volume of coverage is mandatory in routinely performed static and dynamic myocardial perfusion imaging by CT in order to quantify the myocardial blood flow and to evaluate lesion-specific ischemia. Full-service CT (coronary anatomy and ischemia) is the main technique by which to avoid additional functional examinations (i.e., stress echo, MR NMPI, or FFR) in the patient population with moderate CAD.

Despite the fact that descriptive conclusions are still common in CCT reports, most centers in Italy use CAD RADS and the trend is increasing, although CAD-RADS cannot be considered a substitute for the impression provided by the reading physician, who is always consulted to interpret the CCT findings based on the more detailed patient-specific information.

### 1.2. Magnetic Resonance

Cardiovascular Magnetic Resonance (CMR) is the primary imaging modality for non-invasive myocardial tissue characterization, and it is considered the gold standard test for quantifying cardiac function, myocardial volumes, and mass [11]. Since its first applications in the cardiovascular field at the end of the last century, MR imaging has been recognized for its specific diagnostic strength in its ability to characterize myocardial tissue, through non-parametric sequences qualitatively or semi-quantitatively evaluated, specifically for the identification of oedema, fat, iron, amyloid, necrosis, and fibrosis in basal conditions and ischemia in stress settings. Thanks to its possibility of evaluating myocardial tissue by exploiting the magnetic properties of the myocardial structures, it has been increasingly applied in different clinical scenarios, and has had a significant impact on the management of patients with heart involvement [12,13].

In recent years, the introduction of parametric mapping techniques has permitted the evaluation of quantitative changes in the myocardium, based on T1, T2, T2*, and extracellular volume (ECV) parameters, which are linked to tissue characteristics, as proven by bioptic correlation studies [14]. This represents a significant step forward in the quantitative diagnostic capacity of various myocardial pathologies, which in the past often required invasive investigations. The main advantages of a quantitative evaluation are objective differentiation between the pathology and normal conditions, the grading of disease severity, and the possibility of comparing different groups of patients and normal subjects.

Thus, mapping has introduced a new frontier in cardiovascular radiology, enabling the quantification of properties of the myocardium, comparing them with normal reference values acquired under the same scanning conditions [15,16]. Previously, diffuse myocardial disease had always been difficult to measure or even appreciate without invasive procedures. Thus, this advance is of the utmost importance, because focal and diffuse changes often directly reflect the pathophysiologic processes of various diseases from their preclinical phase up to the end stage, and it promises further advancement in the treatment of cardiological patients through better quantitative diagnostics and inter- and intra-patient comparability, allowing patients to be followed over time. Consequently, mapping techniques have the potential to act as biomarkers to help diagnostic decision-making in several pathologies [17], leading to a significant improvement in the prognosis of patients with cardiac diseases by tailoring the therapy. Furthermore, the use of mapping in the context of medical research allows quantitative endpoints to be set in phase 2 and 3 clinical trials without the need for invasive testing.

Initially, much of the research published on parametric imaging focused on progress in acquisition methodology. This achieved numerous, fast, and robust mapping software, commercially available today on state-of-the-art CMR systems. Evidence for the clinical value of CMR mapping from large clinical outcomes trials is rapidly growing. Thus, parametric mapping can be considered a natural extension of comprehensive CMR protocols for deep and quantitative myocardial assessment [14]. Moreover, thanks to technological improvements, it is now possible to analyze the entire left ventricular myocardium with a global or segmental view that shows the intrinsic advantage of higher sensitivity (Figure 3) [15,16,18].

The main limitation of spreading the mapping techniques in the clinical routine and in the research arena is due to its high reliance on the single scanner, the type of sequence used, and on imaging acquisition modality. This is the reason why normal reference values based on sex and age are recommended for each center considering that these parameters can be partially influenced by age and sex [14,19,20] (Figure 3).

The need for a more in-depth characterization of heart function has led, in the last few years, to the development of tissue tracking analysis. This technique provides a quantitative assessment of the global and regional kinetics of the myocardium, (Figure 4) providing adjunctive information about heart function other than that obtained by the gross ejection fraction (EF) alone [21]. In fact, the EF is a parameter of overall ventricular systolic function and does not provide information on regional myocardial kinetics and contractility, nor does it allow for an evaluation of the diastole. Moreover, in many heart diseases, the EF is only altered late in the progression, leaving a diagnostic gap in asymptomatic or subclinical conditions. Since its introduction in the late 1980s, tissue tracking has been extensively developed and is now gaining popularity due to the development of software for fast and robust analysis, as well as due to research trends generating evidence for the adjunctive value of strain in the early and differential diagnosis, risk stratification, and prognostic determination of many heart diseases [21]. Recent evidence has given strength to this technique, and so it is starting to be included in CMR assessment. As for mapping, the determination of normative values in both healthy subjects and the disease-specific population is important to avoid misdiagnosis.

Another promising development in CMR imaging is the quantitative assessment of the functional mean of coronary artery disease (CAD) [22]. Quantitative stress perfusion imaging seems to allow for greater reproducibility and improved diagnostic accuracy compared to the qualitative stress CMR methods, thus improving patient management. The quantitative perfusion technique could be carried out with different protocols, such as dual-bolus, pre-bolus, and single bolus with a dual sequence, and has been proven to be able to detect functionally significant coronary stenosis, with higher cost-effectiveness compared to anatomical assessment with computed tomography coronary angiography [5]. Furthermore, in this case, some considerations should be made, especially regarding the limited availability of quantitative perfusion CMR.

Lastly, another topic worthy of mention is the introduction of four-dimensional (4D) flow MRI, which provides a “time-resolved” form enabling a comprehensive visualization of the blood flow in the heart and throughout the human circulatory system, with the possibility of determining defects and abnormalities [23]. This promising tool is expected to be increasingly applied, providing new possibilities through non-ionizing imaging methods.

Advanced CMR technologies that allow for quantitative evaluations have had a significant impact on clinical management, optimizing diagnostic processes, and evaluating prognostic factors, with the aim of guiding new and personalized therapeutic approaches. The identification of new biomarkers within the context of precision medicine will also lead to evidence of therapeutic efficacy.

The greatest difficulty in optimizing MR use lies precisely in the availability of adequate resources and in technological progress, which requires constant updating, as suggested by international consensus documents. Another aspect of fundamental importance is represented by the need to acquire experience in the sector, to gain specific knowledge in the use and interpretation of the most advanced techniques in the various clinical scenarios.

### 1.3. Future Perspective: National Networking for Big Data and Artificial Intelligence, Cardio-Radiologist in the Heart Team Using Specific Training

Through the technological transfer of innovative quantitative cardiac CT and RM approaches, further developments in efficiency will be possible, as well as through external collaborations among different centers, with bioengineers for cardiac CT and MR image analysis, and hardware and software development, leading to the creation of national and international networks. The formation of a quantitative imaging network as a comparison and communication tool, allowing remote consultation between specialists and the sharing of clinical-diagnostic protocols, may promote new initiatives and side projects, through which research, knowledge, technological innovation, and high-quality care for the patient are produced. Moreover, the development and analysis of large, national high-quality databases would, on behalf of the scientific societies, would provide the basis for the application of artificial intelligence techniques.

Thus, the application of advanced knowledge and skills for research and clinical purposes may implement multidisciplinary projects and collaborations to test innovative precision therapeutic strategies through quantitative cardiac CT and MRI imaging protocols, positioning the cardiac radiologist as a pivotal factor.

A Close interaction among other physicians involved in the management of patients with cardiac disease (cardiologists, cardiac surgeons, reference physicians for systemic disease involving the heart, and nuclear medicine physicians) is crucial because the sharing of specific knowledge and competence is of utmost importance for the choice of management, pharmacology, interventions, and surgery planning [6]. Aligned with the development of multidisciplinary tumor boards, the heart team has become a clinical necessity in optimized patient center-based management.

The involvement of dedicated cardiac radiologists within the multidisciplinary team brings many advantages, including multimodality knowledge, expertise in 2D and 3D image interpretation, detailed knowledge of radiation exposure/-control and safe procedures, and expertise in the interpretation of non-cardiac findings [6].

For the principle of maintaining full respect for each other’s expertise area, in Italy, the radiologist is the unique report’s legal manager (D Lgs 101, 2020 and DM 14 gennaio, 2021). Thus, expanding education and training opportunities, starting with residency, together with certified advanced cardiac life support (ACLS) courses, is the first aim to achieve quality improvement and high standards of clinical service in cardiac imaging in our country.

Just as today’s, cardiac imaging, with its ability to provide diagnostic, prognostic, and risk assessment for tailored therapy, was unimaginable just two decades ago. Dedicated cardiac radiologists with specific training and legal recognition on the model of the path taken in neuroradiology are the main road to further expand the future in cardiac imaging.

## 2. Vascular Imaging

Vascular imaging is an old discipline dating back to the 1950s, when angiography was the preferred method. The development of computed tomography (CT) in the late 1970s progressively superseded angiography, becoming the standard imaging modality by which to assess vascular diseases [24].

With the introduction of ECG-triggered acquisition, the need for additional scans due to pulsation artifacts was prevented, and high accuracy in the imaging of the aorta and nearby vessels was progressively achieved in the 1990s [25].

In Italy, CTA is the technique routinely employed in aortic imaging. Today, multidetector row CT (MDCT) scans are highly efficient, widely available, and highly sensitive and specific in diagnosing acute aortic syndromes (AAS), such as dissections, intramural hematomas, and penetrating ulcers. Moreover, MDCT provides detailed insights into the vessel’s thickness and composition, allowing for the evaluation of vasculitis, and in the emergency setting, it has been increasingly employed to localize the sources of gastrointestinal bleeding [26]. Maximum intensity projection (MIP) reconstruction provides a global assessment and the rapid detection of vascular stenosis and occlusion [27]. Pre-procedural MDCT is crucial for percutaneous interventions, such as transcatheter-aortic-valve-replacement and transcatheter-mitral-valve-replacement. While transthoracic echocardiography is often employed in monitoring post-procedure results with a limited acoustic window, potential complications are generally investigated by CT scans [28]. In routine surgical scenarios, post-processing techniques, such as multiplanar reformats and segmented volume-rendered (VR) reconstructions, have greatly improved the assessment of vascular anatomy and are generally requested by cardiac surgeons (Figure 5).

Although confined to third-level care centers, 3D-printed models have increasingly been used in preoperative settings to improve surgical approaches, particularly in congenital heart disease (CHD). However, standardized protocols are lacking and cardiac valve leaflets cannot be well demonstrated in 3D models [29]. Future advancements include improvements in hardware, computer-aided diagnosis (CAD) software, and functional and physiological imaging. Artificial intelligence and deep-learning-based automated software have increasingly been integrated into CT workstations to reduce operator dependence [30].

New CT systems, such as dual-energy CT (DECT) and photon-counting CT (PCCT), have been recently introduced. DECT surpasses conventional CT scans using attenuation measurements from different energy spectra, quantifying material composition [31]. This novel method facilitates the separation and quantification of iodine concentrations, enabling a better assessment of tissue perfusion, and it is particularly interesting in the study of acute aortic syndrome, where extravasation, intramural hematoma, and endoleak are challenging to accurately detect by conventional CT [32]. PCCT offers complete spectral multi-energy data information, with a small detector pixel design and better spatial resolution facilitating the assessment of iodine density and the quantification of diverse virtual monoenergetic images. PCCT guarantees that patients reduce noise and artifacts along with minimal radiation doses for evaluating disease processes and optimal dose efficiency. This ensures a superior CT angiographic examination and enhances patient safety [33]. Through the intravenous administration of two contrast agents, PCCT can capture endoleak dynamics and discriminate endoleaks from intra-aneurysmatic calcifications in a single scan, reducing radiation exposure. It also has the potential to improve the visualization of the stent deployment [34]. Limitations of PCCT are limited field of view, photon flux and charge trapping [35].

In a non-acute setting, MDCT has been traditionally preferred to magnetic resonance (MRI) because of its time consumption, inferior spatial resolution, and lower availability. However, the assessment of large vessels by Gadolinium-enhanced MRI is exponentially growing in the follow-up of both pediatric and adult populations thanks to the latest advancements in MRI technology due to faster acquisitions and the advantage of being an ionizing-free technique, with a specific diagnostic strength for tissue characterization [36]. Furthermore, MRI can provide additional information on ventricular, valvular, and vascular function and flow dynamics by using cine Steady-State-Free-Precession (SSFP) and phase contrast sequences. Advancements in Fast-Spin-Echo (FSE) black-blood T1, proton density, and Short-Tau-Inversion-Recovery (STIR) T2 sequences have improved the assessment of vessels, making them a valuable tool for aortic wall pathology (Figure 6) [37].

The clinical use of 4D-flow MRI has become avaible only recently. With ECG-gated time-resolved acquisition, a 3D image of blood flow in all three spatial directions can now be captured, allowing for the visualization of the blood flow in the vessels extending from the brain to the lower extremities [23]. Additionally, assessments of wall shear stress, pulse-wave velocity, and pressure gradients can be performed alongside blood flow estimations [38]. MRA combined with 4D flow MRI offers an advantage over dynamic CTA in imaging the whole aorta with no radiation burden; however, its sensitivity in detecting endoleaks has yet to be determined, and further large-scale studies are necessary to determine its optimal use. Additionally, the 4D flow may underestimate WSS due to spatial resolution limitations [39].

Even if 35 years old, Doppler ultrasound remains the least invasive and cost-effective technique in cardiovascular radiology; however, its use is confined to specific anatomical sites. Vector Flow Imaging (VFI), a new technique for angle-independent velocity estimation using ultrasound, has surpassed traditional spectral Doppler ultrasound in terms of accuracy and precision. VFI can grade stenosis by evaluating the flow complexity of the ascending aorta, carotid, and femoral arteries and enables real-time quantification of vortices in any part of the vessel. VFI studies have focused on assessing vortex formation in children with CHD and in adults with straight or complex geometries, and have concluded that the flow is much more intricate than previously assessed by conventional Doppler ultrasound. However, this was a preliminary study in a small population, and integrated fully automatic techniques for quantitative flow assessment are still lacking [40].

Due to the increasing requests of vascular imaging in our country, there is a general approach to provide hospitals with sophisticated machines able to improve spatial and temporal resolution, conspicuity, signal-to-noise ratio, and field of view. Even if the qualitative assessment of vessel anatomy is still the most routine approach, quantitative measurements such as flow complexity, vorticity, perfusion, vessel density, and tortuosity are being increasingly considered. Under these assumptions, the future scenario in our country will be increasingly based on dynamic imaging, such as 4D flow MRI and dynamic CTA, which will provide information regarding vessel anatomy and composition, hemodynamic patterns, and wall shear stress on a large scale. Finally, interesting results have been attained through Computed Tomography Texture Analysis (CTTA). Through texture analysis, it is possible to obtain precise quantitative data from CT scans, which can offer a thorough understanding of the correlation between phenotyping and tissue pathology [41]. CTTA has shown potential for improving the diagnostic accuracy of aortic dissection [42], aortic aneurysms [43], and their assessment after EVAR [44]. Moreover, by quantifying periaortic fat and aortic wall calcifications, TA could be a potential tool for identifying patients at a higher risk of developing cardiovascular diseases [45]. However, Machine Learning algorithms require large databases for learning and training and data sharing is subject to ethical and legal considerations, making the developing of large multicentric registries challenging [45].

In summary, implementing standardized CT protocols across multiple centers and utilizing computer-aided software are crucial steps to improve the visualization quality of vascular systems and reduce radiation dosage and operator variability.

## 3. Rectal Imaging

The latest studies on rectal cancer have mainly focused on the identification of residual disease after neoadjuvant chemo-radiotherapy (nCRT).

This is because a complete or major clinical response to locally advanced rectal cancer (LARC) after nCRT, jointly with the absence of loco-regional nodal metastases, allows patients to be enrolled in rectum-preserving approaches [46].

Since a few years ago, rectum-sparing approaches after nCRT were only reserved for patients that were not fit for classical surgical interventions because of other systemic diseases or for patients that refused surgery. There is a growing body of scientific evidence that supports the feasibility and safety of rectum-sparing strategies in patients that have had a major or complete clinical response to nCRT, since there is a reduction in side effects and acceptable oncological outcomes [46]. Basically, the two main strategies that have been proposed and tested to preserve the organ are watch-and-wait protocols, where surgery is avoided and the patient is strictly monitored with MRI and endoscopy, as well as transanal local excision, where the residual scar/lesion is removed and a histopathology report of the primary tumor can be obtained, thus allowing a quantification of the risk of mesorectal metastases [46]. The latter approach is not free of complications, since it has been reported that in one-third of patients that underwent local excision, a subsequent total mesorectal excision (TME) was required and the bowel function in this group of patients was worse than those who had direct TME [47].

Anyway, these rectum-preserving strategies have been proven to reduce the side effects associated with classical surgical approaches such as TME or the abdominal-perineal resection, granting a better quality of life to the patients [46].

The role of medical imaging in this field is mainly divided into the evaluation of the residual of the local tumor (T staging) and the identification of nodal metastases (N staging).

### 3.1. T Staging

In 2018, the European Society of Gastrointestinal and Abdominal Radiology (ESGAR) published an expert consensus on MRI rectal cancer staging and restaging [48]. The suggestion of ESGAR for the evaluation of local tumors after nCRT was to distinguish: (i) a complete normalization of the rectal wall; (ii) a fibrotic wall thickening without clear residual mass; (iii) residual mass. The normalization of the rectal wall is appreciable in 5% of cases [49]. Some studies have proposed an analog to histopathological tumor regression grade (TRG), which has been named mrTRG, first of all by the Magnetic Resonance Imaging and Rectal Cancer European Equivalence Study (MERCURY) [50] (Figure 7).

With this score, the tumor response to nCRT is divided into five classes from mrTRG1 (no residual tumor signal) to mrTRG5 (no changes in the tumor after chemoradiation); however, it shows a poor correlation with the histopathological evaluation, even if the agreement between radiologist is moderate [50].

Other authors have proposed a morphological evaluation of the local tumor based on T2-weighted sequences. The whole tumor volume after nCRT and the percentage of volume reduction have been shown to be effective in the prediction of a complete response to nCRT, more than one- or two-dimensional measurements, with accuracies of up to 85% [51]. The main disadvantage of this method is that it is time-consuming and challenging in terms of its applicability in a routine clinical setting.

The use of diffusion-weighted imaging (DWI) sequences has also been exploited in the identification of the complete response of the tumor to nCRT. Many studies have underlined how DWI sequences combined with T2-weighted images can help radiologists in detecting the area of the residual tumor after chemo-radiation in the tumoral mass; even when, in a 2019 review by Norvat et al., they were not officially recommended and listed as a controversial practice (“maybes”) [51,52]. Bates et al. proposed the use of a high b value (up to b1500), such as in MRI prostate evaluation, underlying how this helped in the identification of residual rectal cancer after nCRT even if this was tested in a small cohort of patients [53].

The evaluation of the tumoral volume with DWI sequences has been also evaluated, demonstrating good accuracy in complete response detection (Δ volume sensitivity and specificity of around 70–80% and 84–93%), anyway suffering from the same technical difficulties as volume measurement using T2-weighted images [51]. Moreover, the evaluation of the apparent diffusion coefficient (ADC) maps derived from DWI images was tested in complete response to nCRT detection [51]. The best results were obtained in the evaluation of the percentage change in ADC values, more than in the measurement of ADC values before and after nCRT; anyway there were heterogeneities among the cut-off identified by different research groups [51].

The role of dynamic contrast-enhanced (DCE)-MRI with the evaluation of different parameters derived from the contrast in-flow and out-flow curves in the lesion [V_p_ (plasma volume), K_trans_ (plasma-to-extravascular volume transfer), K_ep_ (extravascular-to-plasma volume transfer constant), and V_e_ (extravascular extracellular volume fraction per unit of tissue volume)] is still debated, shown through the conflicting results in the literature [51].

### 3.2. N Staging

A meta-analysis published in 2020 underlined how the accuracy of MRI for nodal staging after nCRT is suboptimal, with a sensitivity of 77%, a specificity of 77%, a positive likelihood ratio of 3.40, and a negative likelihood ratio of 0.30 [54].

Naturally, the presence of loco-regional nodal metastases makes the TME mandatory, since a rectum-sparing approach would put the patient at high risk of local recurrence or distant metastases.

On the other hand, the suspect of loco-regional metastases, not confirmed by histopathology, would expose the patients to over-treatment.

Hence, since the publication of the ESGAR criteria for the MRI staging and restaging of rectal cancer, where the criterion proposed for the identification of loco-regional nodal metastases was the presence of lymph nodes with a short axis greater than or equal to 5 mm [48], different studies have tried to improve the accuracy of MRI for the identification of loco-regional metastases, mainly focusing on a dimensional or morphological evaluation of the lymph nodes.

The morphological criteria that were deemed to be useful at baseline MRI proved to be inaccurate in the restaging setting, with a low agreement between different readers [50,55].

Considering these results, different studies have exclusively focused on the lymph node size to detect nodal metastases.

Heijnen et al. [56] measured all of the lymph nodes visible on a three-dimensional T1-weighted gradient echo sequence in the mesorectal fat of patients affected by LARC after nCRT and identified 2.5 mm of the short-axis as the best cut-off for nodal metastases detection with a sensitivity of 75% and a specificity of 64%, but the study suffered from a limited sample size of 39 patients.

Pomerri et al. [55] evaluated the global mesorectal lymph node reduction rate between staging and restaging MRI (i.e., the reduction of the sum of all of the short axes of the lymph nodes within the mesorectum) and identified a 70% reduction rate as the best cut-off to identify patients without nodal metastases. This method showed a very good negative predictive value (95%) and sensitivity (90%), but it suffered from low specificity; moreover, the measurement of the short axes of all of the mesorectal lymph nodes, both at staging and restaging MRI, is time-consuming and hardly applicable in a routine clinical setting [50,55].

DWI sequences have also been proposed for the detection of nodal metastases within the mesorectum in LARC after nCRT for the detection of nodal metastases [57].

Van Heeswijk et al. [57] showed how the absence of mesorectal lymph nodes in DWI sequences in a post-chemo-radiation setting had a 100% negative predictive value for the differentiation between patients with and without nodal metastases during histopathology, thus suggesting a role for this kind of sequence in the selection of patients that should be addressed for rectum sparing approaches. The main disadvantages of this method are very low specificity (14%) and positive predictive values (24%).

Another issue is the presence of latero-pelvic metastatic lymph nodes (i.e., obturator, internal iliac, or external iliac) as lateral pelvic lymph nodes dissection is not routinely performed in Italy, as recommended by the European Society of Medical Oncology guidelines [58]. Indeed, it is still debated whether this intervention is worth the expense in terms of surgical risks and collateral damages for the patients. In 2022, Kroon HM et al. published a systematic literature review showing that the survival rate of patients treated with lateral lymph nodes dissection was not significantly different from that of those who had had standard TME performed, even if a lower rate of local recurrence was reported [59].

Based on these assumptions, the imaging evaluation of lateral pelvic lymph nodes is pivotal in directing patients toward the best surgical approach.

In 2019, the Lateral Node Study Consortium, through a study including more than 700 patients restaged after nCRT, suggested 4 mm and 6 mm of the short axis as cut-offs for obturator and internal iliac nodes in order to identify metastatic lymph nodes and to direct patients toward lateral pelvic dissection [60].

### 3.3. New Techniques and Applications

Combined [18F] FDG PET/MRI has recently been proposed as an effective imaging modality for rectal cancer patients, capable of generating high-resolution anatomical and functional data. This combined imaging modality can also spare patients the radiation exposure associated with the CT component of PET/CT. PET/MRI achieves a high soft-tissue contrast that is useful for delineating local tumor extent, and it can be implemented with ‘functional’ MR sequences such as diffusion-weighted imaging (DWI) [61].

In a recent meta-analysis, the sensitivity and specificity for T staging, N staging, and M staging in colorectal patients were 95%/79%, 81%/88%, and 97%/93%, respectively [62]. While focusing on rectal cancer, PET/MRI, thanks to its high accuracy in T and N staging, was shown to be a good tool for restaging after nCRT (Figure 8) and to direct patients toward rectum-sparing techniques; while for M staging, there was an intrinsic weakness in terms of lung parenchyma evaluation [63].

Another interesting field is the application of radiomics and texture analysis in the prediction of response to nCRT. Several studies have tested, with promising results, the capability of different textural T2-weighted, DWI, and contrast-enhanced parameters to identify the complete pathological response after nCRT [64]. The prediction models proposed with the radiomics feature showed an area under the ROC curves ranging from 0.71 to 0.93 for complete response identification [64]. Recently, some research groups have proposed models combining both radiomics features and clinical/MRI data, obtaining very good results [65,66]. It is worth underlining that, in certain cases, radiomics analysis of the MR images did not show desirable accuracy for complete response detection [67]. The texture analysis was applied on PET/MR images too, combining both MR and PET textural features, and the accuracy of the reported model was 74%, despite the fact that the sample size of the study was small (50 patients) [68].

In conclusion, the data present in the literature are encouraging, even if there are differences among protocols and models that make the generalizability of the results difficult.

## 4. Liver Imaging

Epidemiological changes in liver diseases, advances in technologies and treatment opportunities, and the pressing need for non-invasive and fast assessment of diffuse and focal liver diseases have had a huge impact on the approach of abdominal radiologists to liver imaging in the last few decades.

### 4.1. Diffuse Liver Diseases

#### 4.1.1. Clinical Setting

Chronic liver diseases are highly prevalent worldwide and develop as a wound-healing response to acute or chronic liver injury. After years of hepatic inflammation triggering progressive fibrosis, cirrhosis may occur, constituting the most important risk factor for hepatocellular carcinoma (HCC). The majority of chronic liver diseases in the developed world include alcoholic liver disease, chronic viral hepatitis, including hepatitis B and C, non-alcoholic fatty liver disease (NAFLD), and hemochromatosis; among these, the prevalence of NAFLD in the general population has significantly increased over the last few decades. NAFLD may evolve into nonalcoholic steatohepatitis (NASH) with the development of inflammation and fibrosis, which may eventually lead to cirrhosis, liver cancer, end-stage liver disease, and death [69].

#### 4.1.2. Imaging Approach

Most patients at risk of chronic liver diseases are seen in primary care. Therefore, given the alarming epidemiological trend and the potential adverse patient outcomes, the adoption of using some non-invasive tests, including ultrasound (US), is suggested in the primary care setting in at-risk groups, primarily those with type 2 diabetes and obesity and those who use alcohol [70]. The NAFLD Consensus Consortium has proposed the use of non-invasive testing for risk-stratifying patients in primary care, for diagnosing and staging non-alcoholic steatohepatitis (NASH) in terms of steatosis and fibrosis burden in secondary care, and has also indicated imaging techniques as important tools [69].

#### 4.1.3. Ultrasound

US-based modalities for the quantification of fat and fibrotic liver content have moderate diagnostic accuracy but may be useful as a screening strategy for their wide availability [71]. Controlled attenuation parameters, attenuation imaging coefficients, sound speed estimations, calibrated US, and US elastography have been proven to be promising for the assessment of steatosis and fibrosis severity [70]. US-based methods are, however, limited in monitoring liver changes over time.

#### 4.1.4. Computed Tomography

The adoption of the multi-material decomposition algorithm in contrast-enhanced dual-energy CT and the use of photon-counting CT (PCCT) seem very promising for liver fat quantification, although these data need further validation [71,72].

#### 4.1.5. Magnetic Resonance Imaging

MRI-based tools, such as MRI-proton density fat fraction (PDFF), MR elastography, or MRI corrected T1, allow for the further risk stratification of NAFLD patients in whom other non-invasive tools are deemed indeterminate or not reflective of clinical suspicion [69,70]. Among MRI tools, fat quantification by MRI-proton density fat fraction is superior to the controlled attenuation parameter and is being increasingly used in tertiary care centers for hepatic steatosis quantification or in clinical trials [69,70,71,73]. MR elastography is considered the most accurate non-invasive method for detecting advanced fibrosis and for differentiating the various stages of fibrosis with moderately high accuracy [71]. However, costs, patient acceptance, and the disadvantage of not being a point-of-care technique still limit the wide adoption of MRI in the primary setting. 

Technological advances in MRI for the study of chronic liver diseases have been also proven for the quantification of hepatic iron concentration, which may occur in hereditary hemochromatosis (i.e., the most prevalent genetic disease in the Caucasian population of North European origin) or secondary hemochromatosis (i.e., Italy is in the black belt of high prevalence for hemoglobinopathies). MRI provides a quantitative, safe, and noninvasive approach to determining the liver iron concentration (LIC). LIC may be calculated using the signal intensity ratio (SIR) and R2- and R2*-based relaxometry methods. However, today, the latter is recognized as the state-of-the-art method for the accurate, validated, reproducible, and fast quantification of LIC [74] at both 1.5 T and 3 T [75].

MRI does not allow the direct measurement of iron, but rather helps to estimate the iron concentration through the effects of iron on the rate of proton signal decay. Therefore, any effect that alters the apparent signal decay rate, such as fat, may introduce a bias for iron estimation. In recent years, methods for simultaneous fat-water separation and R2* mapping have been introduced that allow for unconfounded R2* relaxometry in the presence of fat [74,75], also providing simultaneous quantification of liver fat, which is advantageous in NAFLD (Figure 9).

Therefore, commercially available confounder-corrected R2*-based LIC quantification should be preferred for its accuracy and reproducibility in the quantification of LIC, being also strengthened by the presence of regulatory approval and its rapid acquisition time [76].

#### 4.1.6. Future Imaging Trends

One of the limitations in the adoption of MRI tools is the limited availability of MRI scanners, as well as the need for expert radiologists for interpretation. CT scanners are more widely available, but are limited by exposure to ionizing radiation. However, in the setting of patients with chronic liver disease at high risk of HCC, either CT or MRI are indicated, with CT being more commonly adopted. In addition, many patients undergo CT for other reasons, which may include emergency or oncological settings. As such, postprocessing software has been developed to extract quantitative data from CT scans for liver imaging to quantify steatosis and the hepatic morphologic fibrotic changes occurring in chronic liver diseases. Several promising CT techniques have been proposed for quantifying steatosis, iron, and staging hepatic fibrosis with post-processing software, including radiomics and deep learning [71,77]. Significant research is being published using these novel tools, and many of them still need validation in different populations and using different scanners.

### 4.2. Focal Liver Diseases

#### 4.2.1. Clinical Setting

Focal liver lesions encompass a large spectrum of entities with different pathogenesis, clinical presentations, imaging features, outcomes, and management. Incidentally detected solid liver lesions in healthy patients are mostly benign. The most common benign liver lesions are hepatic cysts and hemangiomas, while focal nodular hyperplasia and hepatocellular adenomas are far less common. Hepatic cysts, hemangiomas, and focal nodular hyperplasia have a benign course and do not require any follow-up [78]. Conversely, hepatocellular adenomas may increase in size over time and develop complications, including hemorrhage and malignancy, and therefore, imaging follow-up is needed and surgery may be indicated in selected cases [78,79].

The pathological subtyping of hepatocellular adenomas was updated in 2017 and the identification of different subtypes is clinically relevant in terms of potential complications that may occur based on the subtype [80].

With regard to the imaging diagnosis of malignancy, metastases are the most common malignancies, with colorectal, pancreatic, lung, or breast cancers usually being the primary tumors. Primary malignancies usually originate in the setting of chronic liver disease with advanced fibrosis or cirrhosis, and most commonly include HCC and intra-hepatic cholangiocarcinoma. HCC accounts for almost 90% of all primary liver cancers in cirrhosis [81]. The diagnosis and management of HCC are complex, due to underlying chronic liver disease, different imaging presentations, and multiple treatment options [81].

#### 4.2.2. Imaging Approach

In noncirrhotic patients, most lesions are benign; hence, lesion characterization should be performed at a minimum cost and with high specificity to avoid unnecessary treatment. Imaging diagnosis of the most benign focal liver lesions is oftentimes straightforward through US, CT, and MRI. However, in case of an atypical presentation or uncommon evolutions of benign liver lesions, current imaging techniques may not be considered definitive and further work-up, sometimes including biopsy, may be needed [82]. With regard to malignancies, HCC is the only neoplasm that can be confidently diagnosed through imaging without the need for biopsy confirmation whenever specific imaging criteria are present on contrast-enhanced CT or MRI in patients at high risk of HCC [81]. Therefore, current gaps in knowledge include the diagnosis of HCC when specific imaging criteria are not present or outside the context of at-risk patients and the imaging diagnosis of non-HCC malignancies, in order to reduce the need for biopsy for characterization.

#### 4.2.3. Ultrasound

US elastography, contrast-enhanced US (CEUS), and new Doppler techniques have recently provided some tips as additional US tools to differentiate between benign and malignant lesions, as well as for monitoring after locoregional therapies.

CEUS is currently recommended as the first-line imaging technique for the characterization of incidentally detected focal liver lesions, indeterminate through US in non-cirrhotic, non-oncological patients and in patients with renal insufficiency, as well as a helpful tool in patients with inconclusive findings through CT or MRI [83]. CEUS is also recommended for differentiation between benign and malignant portal vein thrombosis [83], as differentiation may sometimes be complicated through CT or MRI 113, but also in the evaluation of the treatment effect after ablation, guidance for immediate US-guided re-treatment of residual tumors, and in the follow-up after ablation treatment at appropriate time intervals [83,84].

With regard to Doppler techniques, relatively recent advances in Doppler technology have seen the development of advanced Doppler US techniques, including microvascular or microflow imaging (MVI/MFI) and superb microvascular imaging (SMI) [85]. Today, MVI/MFI permits a high-resolution assessment of “microvessel” architecture to approximately 0.5 mm and speeds < 0/1 cm/s without the need for intravenous contrast medium [85]. Benign and malignant focal liver lesions have a different vasculature appearance on MVI/MFI and SMI, which may help in their differentiation; however, the scientific literature on the role of these advanced Doppler US techniques for focal liver lesions is currently limited [85].

#### 4.2.4. Computed Tomography

The adoption of dual-energy CT (DECT) for liver imaging has shown some value in selected cases, particularly for virtual monochromatic images (VMC) and material-specific images, namely iodine images, and with deep-learning reconstruction techniques applied to DECT images [71]. Low keV VMC images and iodine maps can improve the conspicuity of liver lesions and may serve as imaging biomarkers of tumor treatment response [86,87,88]. PCCT may also provide additional help in improving diagnostic accuracy for liver lesion characterization, although, until now, studies have been performed in small cohorts and this technology is not yet widely available [89].

#### 4.2.5. Magnetic Resonance Imaging

MRI techniques have steadily improved over time in terms of their temporal resolution and noise reduction. In addition, abbreviated MRI strategies have been investigated for oncological and HCC screening in cirrhotic patients [71]. All of these efforts are aimed at allowing for wider routine use of MRI as an alternative to CT, given the higher accuracy for focal liver lesion characterization.

#### 4.2.6. Future Imaging Trends

Similar to the current need for quantitative data in the diagnosis of diffuse liver diseases, postprocessing software has been developed to extract quantitative data from CT and MRI scans for liver imaging to differentiate between benign and malignant lesions in the overall population, and to timely and accurately diagnose HCC in the cirrhotic population. On one hand, perfusion images have been obtained both through CT and MRI with the aim of calculating quantitative parameters that might help in the differential diagnosis of focal liver lesions and the assessment of tumor response [90].

A growing number of studies have also been published on the use of radiomics applied to CT and MRI studies for the differential diagnosis of benign and malignant liver lesions, as well as to improve the accuracy of the diagnosis of malignant lesions; however, the clinical routine adoption is still limited.

An interesting potential opportunity still under initial investigation is the possibility of simultaneous biphasic liver imaging in a single multi-energy PCD-CT acquisition using a dual-contrast injection protocol with both an iodine-based and a gadolinium-based contrast; this would have a clear significant benefit for lesion characterization to obtain functional and quantitative information from both contrast agents at the same time [91].

Conclusively, this review highlights the state-of-the-art practices in medical radiology in Italy, specifically focusing on cardiovascular and abdominal imaging. It emphasizes the importance of the latest advancements in US, CT, and MRI technologies, such as “dual-energy” and “photon-counting” techniques, mapping, strain, and 4D flow MRI and hybrid [18F] FDG PET-MRI. Predicting the future is challenging, especially in a rapidly evolving scenario where the last generation of technologies is converging. Radiologists will need a constant updating through education and training to improve image acquisition protocols, to reduce radiation dose and to improve the quality of reports. Such improving of technologies in radiology is supposed to lead to high quality and tailored patient’s care. In this way, a close interaction among the other physicians involved in the management of patients is crucial in optimized patient center-based management.

Moreover, the development of large national high-quality databases, hopefully on behalf of the scientific societies, is recommended for artificial intelligence applications aimed at research and clinical precision medicine.

## Figures and Tables

**Figure 1 diagnostics-13-02439-f001:**
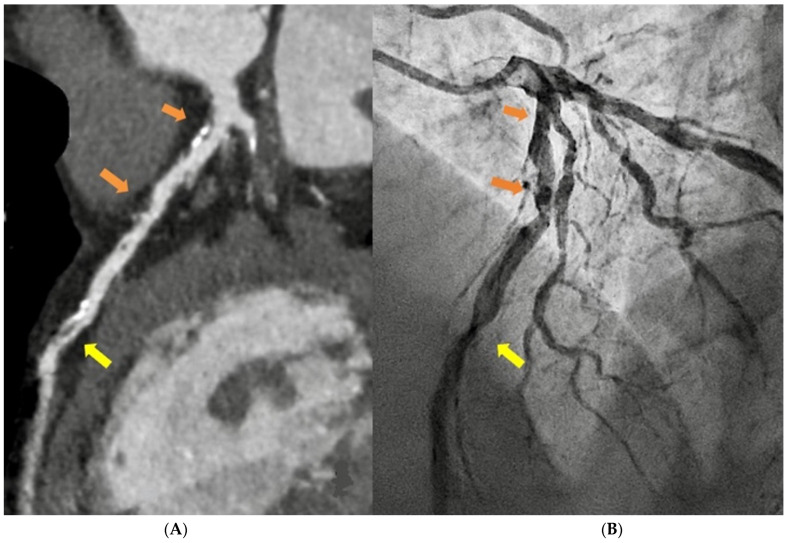
MDCT curved multiplanar reformat reconstruction of the left anterior descending artery (**A**) and respective coronary angiography (**B**) in a 52-year-old male, with positive exercise stress test for inducible ischemia, scheduled for revascularization, showing diffuse calcific and non-calcific plaques with mild stenosis (orange arrow—(**A**)) and severe stenosis with markers of instability (yellow arrow—(**A**)), the same findings were confirmed at coronary angiography (**B**).

**Figure 2 diagnostics-13-02439-f002:**
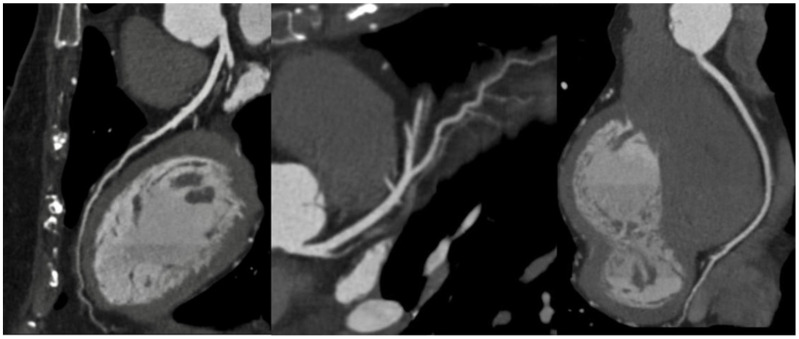
Curved multiplanar reformat reconstruction of the left anterior descending artery (**left**), left circumflex (**middle**), and right coronary artery (**right**), showing patent coronary arteries without stenosis in a 46-year-old female with hypertension and familiarity with coronary artery disease complaining of chest pain with an EKG doubtful for ischemia in the emergency room.

**Figure 3 diagnostics-13-02439-f003:**
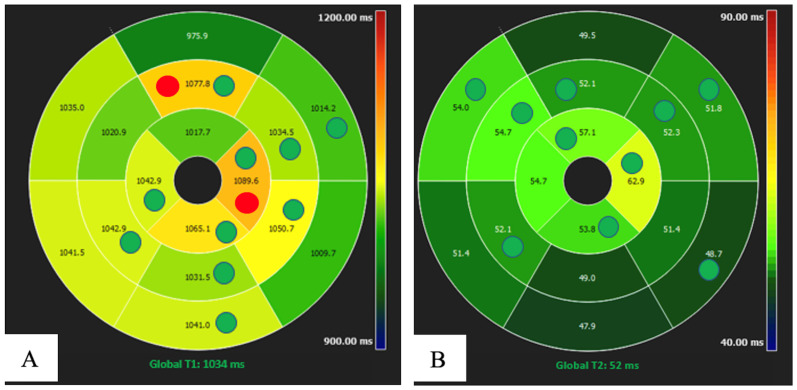
M, 51 years old, with negative cardiovascular history, underwent CMR 3 months after COVID-19 pneumonia. T1 (**A**) and T2 (**B**) mapping global values turned out to be high considering our ranges, but were normal according to the literature. The diagnosis of acute/subacute inflammation at non ischemic pattern was, therefore, possible, and was also confirmed during the segmental evaluation (the green and red circles identify pathological segmental values considering the ranges of our institute and those in the literature, respectively).

**Figure 4 diagnostics-13-02439-f004:**
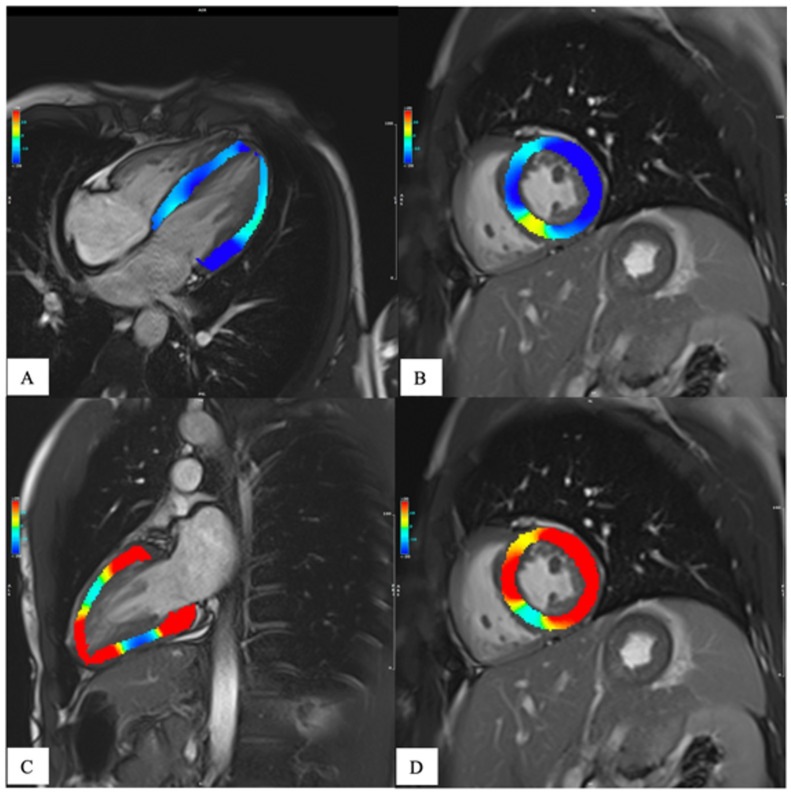
Patient with normal ejection fraction (quantitative evaluation 60%) and altered global strain values, compared to references obtained in our center from 50 healthy volunteers. Representation of longitudinal strain in (**A**) (−19.0%, [n.v. −15.53 ± 1.5]), circumferential strain on middle-short axis view in (**B**) (−19.6% [n.v. −16.3 ± 1.8]), radial long axis strain in (**C**) (37.6% [n.v. 26.53 ± 3.8]), and radial short axis strain on middle-short axis view in (**D**) (35.2% [n.v. 25.75 ± 4.1]), all in systolic phase.

**Figure 5 diagnostics-13-02439-f005:**
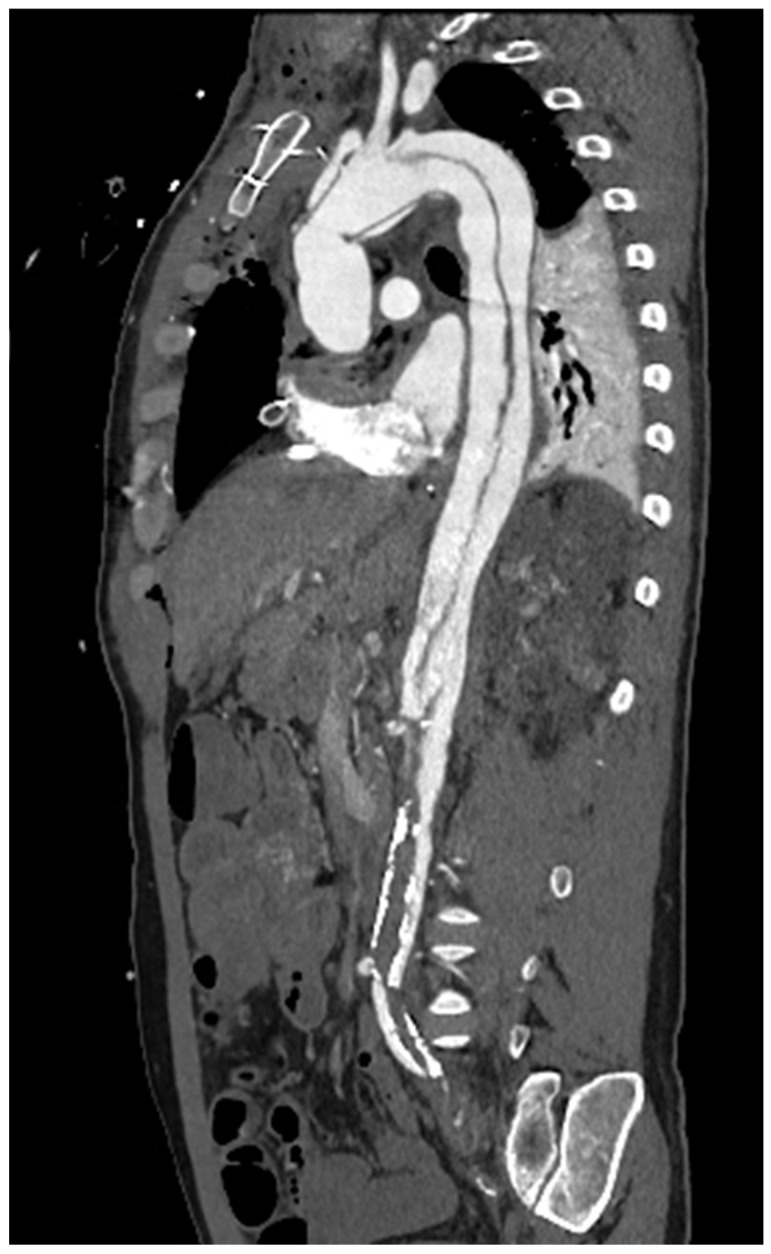
Parasagittal reconstruction status post-Bentall procedure in a patient with type A aortic dissection (sec Stanford) with thrombosis of the true lumen, caudal to the superior mesenteric artery.

**Figure 6 diagnostics-13-02439-f006:**
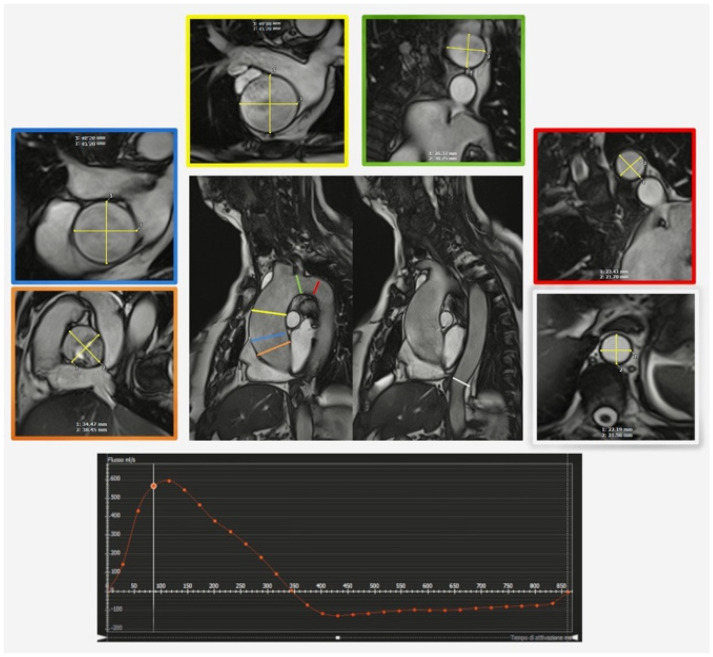
Cine SSFP sequences of parasagittal thoracic aorta in a patient with aortic regurgitation in the bicuspid valve with para–axial sections orthogonal to the long axis of the vessel at aortic sinuses (orange), sinustubular junction (blue), tubular ascending aorta (yellow), medium transverse arch in bovine trunk (green), isthmus (red), and diaphragmatic aorta (white). The flowmetric curve in aortic sinuses by 2D phase contrast (below).

**Figure 7 diagnostics-13-02439-f007:**
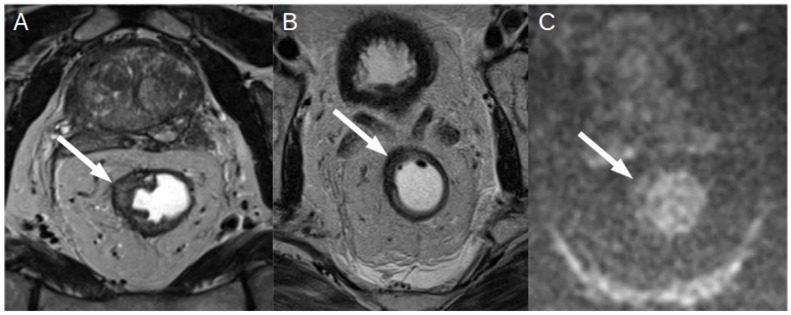
(**A**) T2-weighted axial oblique MRI of T3b rectal cancer (white arrow—(**A**)) before chemoradiotherapy. (**B**) After neoadjuvant therapy, a reduction in the rectal tumor can be appreciated (white arrow—(**B**)) with fibrotic hypointense T2-weighted areas in the lesion but (**C**) with signs of signal restriction in high b value DWI sequences (white arrow—(**C**)). The MRI was scored as T2 mrTRG3 (more than 50% of intra-tumoral fibrosis), and it was confirmed at histopathology.

**Figure 8 diagnostics-13-02439-f008:**
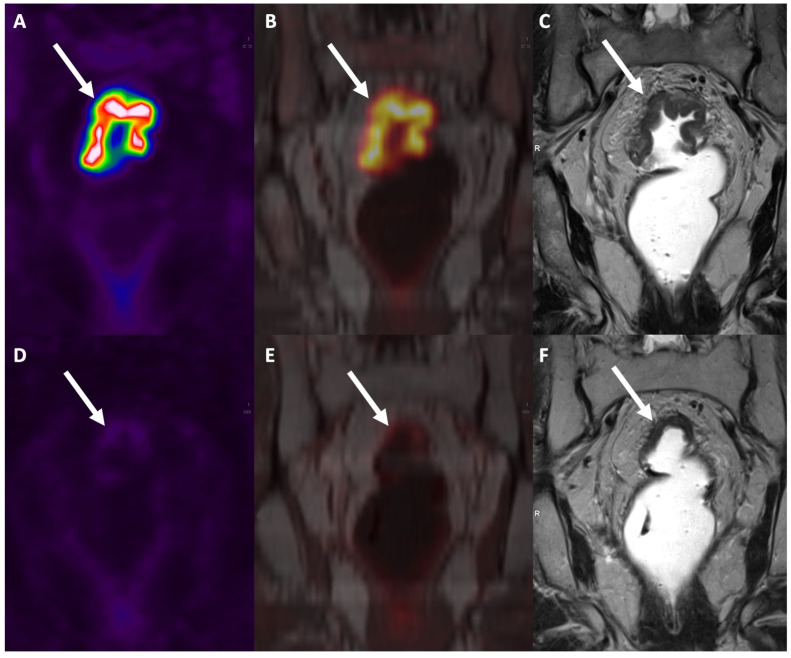
PET/MRI staging of T3 rectal cancer (arrows) shows hypermetabolism in both the PET image (**A**) and the fused PET/MR image (**B**), as well as a rectal wall thickening in the MR image (**C**). Restaging PET/MRI after chemo-radiotherapy shows no hypermetabolism on PET (**D**) and fused PET/MR images (**E**), and a T2 hypointense thickening of the rectal wall on MR image (**F**). The lesion was scored as a complete response (ycT0N0) and confirmed at histopathology after transanal local excision.

**Figure 9 diagnostics-13-02439-f009:**
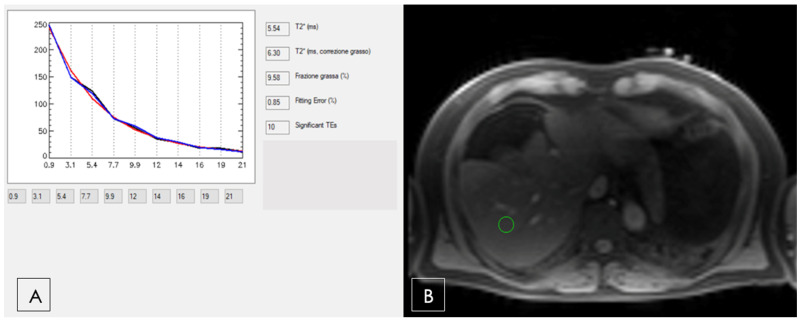
Patient in the lung transplant check-list for idiopathic lung fibrosis. (**A**). The T2*curve demonstrates mild steatosis with a fat fraction of 9.58% and hyperferritinemia (>6000 mcg/mL). (**B**). T2* multiecho BB sequence with TE = 0.93 ms with the green ROI used to develop the curve in (**A**).

## Data Availability

Data sharing is not applicable to this article.

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
