# Peer review of "Medical Radiology: Current Progress"

_diagnostics, 2023, doi:10.3390/diagnostics13142439_

Round 1

Reviewer 1 Report

The article "Medical Radiology in Italy: Current Progress," is a very interesting review of advances in cardiac, hepatic, and oncologic imaging. The review includes the main imaging techniques such as CT, MR, US, and PET/MRI. However, the manuscript requires a detailed review of English writing and grammar. A review on this topic is always relevant and interesting, especially with the inclusion of the topic of the application of artificial intelligence in the analysis of medical images, as the authors emphasize. Therefore, I suggest removing the phrase "in Italy" from the title and simply rewriting the title as "Medical Radiology: Current progress".

The article "Medical Radiology in Italy: Current Progress," is a very interesting review of advances in cardiac, hepatic, and oncologic imaging. The review includes the main imaging techniques such as CT, MR, US, and PET/MRI. However, the manuscript requires a detailed review of English writing and grammar. A review on this topic is always relevant and interesting, especially with the inclusion of the topic of the application of artificial intelligence in the analysis of medical images, as the authors emphasize. Therefore, I suggest removing the phrase "in Italy" from the title and simply rewriting the title as "Medical Radiology: Current progress".

Author Response

Dear Reviewer, we have provided all the modifications you have suggested.

Reviewer 2 Report

The manuscript is fluent and clear, the bibliographic research is wide, the discussion is well conducted.

I enjoyed reading the manuscript and I do recommend publication pending the these minors’ corrections:

-I think it is important to emphasize the possible bias of the studies.

-I miss role of combined [18F]FDG PET/MRI or PET/CT on liver imaging

-I also miss a conclusion.

Minor editing of English language required

Author Response

(The authors gave the same response as above.)
